evolution, plant science

Bateman gradient, polyandry, male–male competition, mating system, sessile organisms

**Author for correspondence:**
Jeanne Tonnabel
e-mail: jeanne.tonnabel@unil.ch

# Do metrics of sexual selection conform to Bateman's principles in a wind-pollinated plant?

Jeanne Tonnabel[1], Patrice David[2] and John R. Pannell[1]

[1]Department of Ecology and Evolution, University of Lausanne, CH-1015 Lausanne, Switzerland
[2]Centre d'Ecologie Fonctionnelle et Evolutive (CEFE), UMR 5175, CNRS, UM, Université Paul Valéry Montpellier, EPHE, 1919 route de Mende, 34293 Montpellier Cedex 5, France

JT, 0000-0003-3461-6965; JRP, 0000-0002-0098-7074

Bateman's principles posit that male fitness varies more, and relies more on mate acquisition, than female fitness. While Bateman's principles should apply to any organism producing gametes of variable sizes, their application to plants is potentially complicated by the high levels of polyandry suspected for plants, and by variation in the spatial distribution of prospective mates. Here we quantify the intensity of sexual selection by classical Bateman metrics using two common gardens of the wind-pollinated dioecious plant *Mercurialis annua*. Consistent with Bateman's principles, males displayed significantly positive Bateman gradients (a regression of fitness on mate number), whereas the reproductive success of females was independent of their ability to access mates. A large part of male fitness was explained by their mate number, which in turn was associated with males' abilities to disperse pollen. Our results suggest that sexual selection can act in plant species in much the same way as in many animals, increasing the number of mates through traits that promote pollen dispersal.

## 1. Introduction

Darwin [1] introduced the notion of sexual selection, recognizing the tendency of males to compete for access to females, and of females to choose their male partners. Bateman (1948) helpfully developed this notion in three basic principles [2] that can be examined by estimating individuals' reproductive and mating success, defined, respectively, as the number of offspring produced and the number of mates. Bateman's principles [2] state that males should exhibit stronger variance than females in (1) reproductive success and (2) mating success, and that (3) reproductive success should depend on mating success in males more than in females. Noting the higher cost of producing female versus male gametes (i.e. anisogamy), Bateman reasoned that male reproductive success should be limited by their mating and fertilization success rather than by investment in each gamete. By contrast, female reproductive success should depend on their ability to produce viable ovules and seeds rather than on the probability of having ovules fertilized [2].

Numerous studies have tested Bateman's principles in animals and, despite some disagreement [3,4], their utility and generality are widely accepted [5]. Male reproductive success relied on mates more than that of females in many animals, as expected by Bateman (1948). Variance in reproductive success also tends to be larger for males than females [6,7], particularly when females care for their young after fertilization, or when males express more elaborated traits [8,9]. Counter-examples have been found where both sexes are similarly energy-limited or mate-limited [10], or in the case of sex-role reversals or female-biased sex-ratios [5,11]. In contrast to animals, the application of Bateman's principles to plants has been limited [12], despite wide acceptance of the role of sexual selection in plant evolution [13–16].

Sexual selection in plants likely occurs through between-male competition to fertilize a limited pool of ovules and may consequently affect the evolution of traits involved in pollen production, export and competitiveness [13–15]. Such potentially sexually selected traits include large flowers and floral displays that enhance pollinator attraction [15,17–19]; increased pollen production [20,21]; male flowering phenology that tracks that of females [22]; vegetative architectures that enhance pollen dispersal [21]; evolution of horn weapons that prevent the attachment of pollen-bearing structure from additional males [23] and high pollen-grain performance [24]. Floral strategies that affect the distribution of pollen on pollinators' bodies could also be under sexual selection [25]. While botanists have commonly described plant female and male functions as limited, respectively, by access to resources and pollinator visits [26], we still know little about the relationship between a plant's mate and its reproductive success (but see [12] for an example in a bryophyte species).

The paucity of attempts to estimate Bateman gradients in plants may be attributable to difficulties in its use, including those that apply generally, and those specific to plants. It is indeed typically difficult to estimate mating success directly. In animals, only a few studies have actually counted mating events, and a proxy for mating success is typically assessed using genetically based paternity assignment in a small subset of the total progeny produced [27]. The number of individuals in the population that share at least one offspring with a given focal individual is classically estimated using the output of paternity assignments [5]. This estimate is thus a genetically based proxy for mating success rather than a direct estimate of mating success itself (hereafter termed mating success proxy). Pollen tracking is possible but remains logistically difficult [25,28], so that such a proxy remains a useful substitute for evaluating Bateman's principles.

The mating and growth habits of plants pose additional specific challenges to the evaluation of Bateman's principles. First, plants are often assumed to be highly polyandrous [29] potentially resulting in extreme mate numbers. Given that genetic assays are made on a finite number of seeds per plant, mates with small contributions to total reproductive success will likely be missed by a genetically based proxy. This could lead to underestimation of variance in mating success and a potential bias towards more positive male Bateman gradients because both the reproductive and the mating success of males are estimated using the same genetic data [27]. Moreover, a genetically based proxy registers mating success only if a male's paternity share exceeds a certain detection threshold (typically determined by the number of seeds sampled per female). This may be a problem when polyandry is high and when most males have a small share in paternity—a situation often perceived as common in plants [29]. If, on the other hand, male contributions are very unequal, focusing on major pollen donors poses less of a problem.

Second, plants are modular, and different flowers on a plant represent separate arenas for competition between pollen donors. Arnold's [30] original definition of plant mating was centred on access to mates. However, studies of plant reproduction have rather defined it as the realized access by an individual to flowers or ovules rather than to mates [15], similar to definitions adopted for aquatic animals with external fertilization [31]. In cases where pollen export from different flowers is largely independent, mating success at the flower level might be a more relevant variable for

sexual selection than that at a plant's level. However, it is also conceivable that a plant's traits (e.g. [17,21,32]) may influence mating success at the plant level, rendering estimates of mate number at the plant level potentially useful.

Plant size and plant architecture provide good examples of traits that may influence pollen production and/or its dispersal distance. Consequently, plant size and architecture may be selected for either through fecundity selection or sexual selection. On the one hand, fecundity selection may select for larger plants that enjoy larger pools of resources that can be allocated to gamete production; this has been termed a 'budget effect' of plant traits [33]. High pollen production may also allow the competitive exclusion of pollen from other males, for instance, by saturating stigmas with pollen [33]. On the other hand, sexual selection may occur through the placement of flowers on elongated branches or inflorescences that favour pollen dispersal, especially in wind-pollinated plants; this has been termed a 'direct effect' of plant traits [21,33]. Analogous ideas have been proposed for animals with a sessile life-form and external fertilization, e.g. in broadcast spawners, in which increased sperm speed and longevity allowed greater siring success over a larger spatial area [34]. Interestingly, the positive male Bateman gradient found at the gametophytic stage of a moss species was achieved through increased clonal growth, and therefore increased individual spatial range [12].

Because plants are sessile, mating patterns are likely to be strongly affected by the spatial location of individual plants and their prospective mates, and thus by density. For instance, males that effectively sire seeds over multiple females may reap benefits associated with reduced local mate (or local pollen) competition (because its pollen grains should compete less intensively with one another; [35]) and local resource competition (because the seeds it sires are less likely to compete with one another both for resources supplied by the female and for resources from the environment during establishment; [33]). By contrast, dispersing pollen over greater distances may come at a cost of pollen dilution, with a correspondingly lower paternity share on female mates nearby. Analysis of Bateman gradients and variance partitioning at the scale of nearby males versus more distantly related mates may therefore illuminate how selection operates on pollen production, pollen dispersal and the resulting relationship between mate number and paternity share in a spatial context.

Here, we consider the utility of Bateman gradients for understanding how sexual selection might operate in a wind-pollinated herb. We conducted paternity analyses based on microsatellite data on the outcome of mating in two semi-natural common gardens of the dioecious plant *Mercurialis annua* that represent extremes in the range of plant densities found in natural populations. We tested Bateman's predictions by calculating: (1) the opportunity for selection capturing variance in reproductive success, (2) the opportunity for sexual selection expressed by variance in mating success and (3) the strength of sexual selection estimated using Bateman gradients quantified as the slope of a regression of reproductive success on mating success. Because variance in reproductive success may vary across stages of the life cycle, we decomposed variance in male reproductive success into an ability (1) to access mates, (2) to secure paternity on their mates and (3) to mate with females with more ovules [36]. We specifically investigated

'direct' and 'budget' effects, assessed by pollen production and dispersal on all fitness components. Because the two common gardens differed in terms of their pot density, we used computer simulations of plant mating in a spatial context to test the hypothesis that the effect of plant density on sexual selection might depend on the scale of pollen dispersal. We tested the hypotheses that larger pollen dispersal distances and, to a lesser extent, higher pollen production could give males greater access to more mates. Finally, we adopted a paired design whereby males and females were grown together in a pot to test the hypothesis that selection for mate acquisition ought to be stronger for access to prospective mates placed further than the immediate surroundings of a focal plant.

## 2. Material and methods

### (a) Study species

*Mercurialis annua* is a wind-pollinated annual herb inhabiting disturbed habitats in western Europe and around the Mediterranean Basin [37]. Populations vary in their sexual system across the species' range, from dioecy, through androdioecy to monoecy [37]. Here, we focused on dioecious populations. Males produce green staminate flowers held on erect inflorescence stalks (peduncles), whereas females produce green dehiscent subsessile capsules in their leaf axils. In both sexes, flowering begins several weeks after seeds germinate and continues over a period of three to four months [37].

### (b) Experimental design

Our study is based on a recently published dataset that estimated male fitness through marker-assisted paternity analyses in two common gardens [21]. Seeds were collected from 35 populations located in northern Spain that were bulked and grown for three generations in a common garden in Lausanne. Male and female fitness components were assessed after mating in two common gardens at varying densities and equal sex-ratios in Montpellier. A peculiarity of the design is that males and females were grown in pairs, allowing us to compare male strategies that were successful at siring ovules locally versus over longer distances.

In each garden, female–male pairs were transplanted into pots that were assigned randomly to a position in a $10 \times 10$ grid. Pots in both gardens were initially established at a low density of 1.0 m between pots. When plants had begun producing male and female flowers, we moved their pots to establish two contrasting densities. In one garden, pots were moved such that the new pot spacing measured 20 cm, while in the other garden, pots were maintained at the same spacing. We constrained plants from both gardens to grow at the same low density initially because we wished to minimize variance in plant architectural traits that might be affected by a plastic response of shade avoidance classical of the high-density population [21]. Plants in both gardens were allowed to continue mating for an additional four weeks, so that all seeds sampled at the end of the experiment had been fertilized under the conditions after pots had been moved (in *M. annua*, seeds are dispersed about two weeks after fertilization, so that seeds sired prior to the change in imposed densities were not sampled). Note that our design does not allow a statistical comparison between densities; we explored the effect of density specifically by means of computer simulations (see below).

In both gardens, leaves of all adults were sampled at the end of the experiment and preserved in silica gel for later DNA extraction and genotyping. All seeds of all 100 females were harvested in both gardens by drying vegetative parts, threshing and winnowing seeds from the samples. Seeds were then counted for each female using an automatic seed counter (Elmor C3; Elmor Angewandte Elektronik, Schwyz, Switzerland). On each male, inflorescences were harvested, dried and weighed to estimate inflorescence weight, which is known to provide a reliable estimate of pollen production [37]. To characterize male dispersal abilities, we extracted individual mean dispersal distances of pollen from previous inferences [21], in which genotype and spatial data were used in a spatially explicit model of pollen dispersal kernels with a negative exponential power function.

### (c) Paternity assignment: estimation of reproductive and mating success

A paternity analysis was performed in each garden separately, based on the genotyping of all adults and 651 and 621 offspring in the low- and high-density gardens, respectively [21]. Genotyping was performed on eight microsatellites [38]. The two paternity analyses were performed using CERVUS version 2.0 [39], allowing for a maximum of four mismatches and accounting for a 0.7% error rate in genotyping. This error rate was calculated as a mean across markers, for both gardens combined, of the overall proportion of offspring whose genotype did not match that of their mother. We assigned paternity based on a 95% confidence (strict) criterion [39]. In the low- and high-density gardens, respectively, 96 and 93 males were assigned as the father of at least one seed, four and seven males were not, and none of the males were excluded from the analysis because of a failure in genotyping.

In females, reproductive success $RS_f$ was estimated as the number of seeds. We calculated male reproductive success $RS_m$ as the sum over all female partners of the product between the proportion of the female's seeds sired by the focal male (estimated best father) and the $RS_f$ of the female partner. Our proxy for mating success for females and males ($MS_f$ and $MS_m$, respectively) was calculated as the number of genetic partners (i.e. the number of individuals in the population that share at least one offspring with a given focal individual). This measure is an estimate of effective mating success given that the probability of detection of a mate is proportional to the number of seeds effectively sired.

The germination probability for seeds of a given mother was calculated based on an average of 10.8 ($\pm 2.29$ s.d.) and 11.3 ($\pm 2.72$ s.d.) seeds sown per female for the low- and high-density gardens, respectively [21]. In females, the number of mates necessarily depends on seed germination rates, since mate number was determined by evaluating paternities of seedlings resulting from the germination trial. For the purpose of consistency between sexes, we presented results using $RS_m$ estimated without weighing the number of seeds by germination probability in both sexes. However, our results were robust when $RS_m$ was estimated by weighing $RS_f$ by seed germination probabilities, accounting for a more integrative measure of male fitness. Distance to the centre of each garden did not affect either mating or reproductive success.

### (d) Quantification of sexual selection

We quantified the strength of sexual selection separately for each sex and assessed the extent of differences between the sexes using: (i) the standardized variance in reproductive success, $I$, i.e. the 'opportunity for selection', (ii) the standardized variance in mating success, $I_S$, i.e. the 'opportunity for sexual selection'; and (iii) the slope of a least-square regression of reproductive success against mating success, $\beta_{SS}$, i.e. the 'Bateman gradient' [40]. These metrics quantify the maximum strength of selection on offspring production ($I$), on selection on mating success ($I_S$), and the fitness gain for one sex for mating with another individual ($\beta_{SS}$). To compare Bateman gradients ($\beta_{SS}$) between sexes we

standardized both mating and reproductive success proxies by dividing by their mean values prior to $\beta_{SS}$ estimation. Similarly, we standardized both the opportunity for selection ($I$) and for sexual selection ($I_S$) by dividing the variance in both mating and reproductive success by the square of their mean value.

Measurement errors on reproductive success typically differ between sexes [41]. Our estimate of $RS_f$ involved direct counts of seeds, whereas that of $RS_m$ relied on estimates of paternity share of a subset (typically $N = 4.8$) of the seeds produced by each female, effectively introducing an additional binomial error component for male compared to female components. Following [41], we computed the expected additional error variance due to binomial errors in males and subtracted it from raw variances to arrive at an estimate with a comparable contribution of measurement error in both sexes, and to be able to compare variances between sexes.

## (e) Decomposition of male reproductive success

We decomposed $RS_m$ into its different components by adapting previous methodology [42,43] to study which fitness components contribute most to variance in male fitness (electronic supplementary material, methods S1). $RS_m$ was first decomposed into intra-pair and extra-pair components, and we obtained a total of six components of variance: (I) the proxy for intra-pair mating success, (II) the paternity share on intra-pair female partners, (III) the fecundities of intra-pair female partners, (IV) the proxy for extra-pair mating success, (V) the paternity share on the extra-pair female partners and (VI) the fecundity of extra-pair female partners. All associated covariances were also assessed. As paternity share and fecundity of female partners cannot be calculated when there are no female mates, we considered only males with MS > 0 to compute their variances and covariances (electronic supplementary material, methods S1); in the case of intra-pair components, males all had MS = 1, so the covariances between MS and paternity share or mate fecundity were undefined. We represented graphically the proportion of variance in $RS_m$ that is attributable to each of these six fitness components and their covariances.

## (f) Statistical analyses

We estimated Bateman gradients ($\beta_{SS}$) by regressing reproductive success against the proxy for mating success at the global, intra-pair and extra-pair scales. Following previous recommendation [27,30], we compared the likelihood of linear and quadratic relationships between relative mating success and relative reproductive success using likelihood ratio tests. Because quadratic components were not significant, only linear regressions are reported. We examined the difference in the strength of sexual selection between sexes by assessing the significance of the interaction between the proxy for mating success and sex.

We estimated the linear relationship between the proxy for relative mating success and both mean pollen dispersal distance and pollen weight (standardized within gardens) using bivariate regression to account for their correlation. We regressed components of $RS_m$ against pollen dispersal distance and pollen weight using either linear models or generalized linear models, depending on the distribution of the fitness component. Both intra-pair mating success and paternity share were analysed using a binomial error distribution, and we accounted for a Poisson error distribution for extra-pair mating success. Generalized linear mixed models were performed treating individuals as random effects to correct for residual over-dispersion (when necessary). Correlations between pollen dispersal distance and weight were examined using Pearson correlation tests. Significance of all the effects described above was examined using likelihood ratio tests. We assessed confidence in variance in $RS_m$, $MS_m$ and in all components of $RS_m$ by performing 10 000 bootstrap samples for all

statistics described in electronic supplementary material, methods S1. We further calculated and plotted 95% confidence intervals for all components of reproductive success (I–XI) and compared the confidence intervals between sexes. We used this bootstrap resampling to assess the significance of covariance between components of male fitness by computing the $p$-value associated with a null covariance in the bootstrap distribution. Finally, we performed bivariate linear regressions for all our fitness components against both pollen dispersal distance and weight to quantify the variance for each fitness component explained by these two traits. All statistical analyses were performed using the lm, glm and glmer functions in the lme4 package [44] in R v. 3.2.2 [45].

## (g) Simulation model of pollen dispersal abilities

The effect of plant density on the intensity of sexual selection was investigated by modelling pollen dispersal from male pollen donors to female recipients (electronic supplementary material, methods S2). Pollen dispersal from each male donor was simulated using a negative exponential function. We calculated simulated $RS_m$ and $MS_m$, and resulting Bateman metrics, based on the males' contribution to the pollen cloud of each female by simulating a sample of eight seeds per female. We compared Bateman metrics calculated in three simulated spatial scenarios with: (1) no variance in pollen dispersal between males; (2) among-male variance in pollen dispersal abilities with a long average dispersal distance relative to inter-individual distances; and (3) among-male variance in pollen dispersal with a short average pollen dispersal distance relative to inter-individual distances. These three scenarios were simulated for both a regular grid (corresponding to our design) and a random distribution of 100 males and 100 females in a squared population.

# 3. Results

## (a) Males and females differed in their Bateman metrics

Both the opportunity for selection ($I$) and the opportunity for sexual selection ($I_S$) were higher in males than in females, regardless of plant density, and none of the bootstrap confidence intervals overlapped between sexes (table 1). In females, no significant relationship was found between the proxies for mating and reproductive success ($\beta_{SS}$), whereas males displayed a significantly positive Bateman gradient (figure 1). In both gardens, such differences were revealed by a significant interaction between sex and the proxy for mating success (mating success × male at low-density: $\beta_{SS} = 1.52$, d.f. = 1, $p < 0.0001$; high-density: $\beta_{SS} = 1.14$, d.f. = 1, $p < 0.0001$; figure 1). In the low-density garden, females displayed a marginally significant negative Bateman gradient (figure 1). Reproductive success was positively related to the proxy for mating success at both the intra- and extra-pair scales (but with a marginally significant effect at the intra-pair scale in the low-density garden; electronic supplementary material, figure S1).

## (b) Male mating success explained substantial variance in reproductive success

Our paternity analysis found an average of 4.97 and 4.62 male partners per female in the high- and low-density gardens, respectively. Local male partners sired a proportion of 0.22 and 0.38 intra-paired seeds at the high- and low-density gardens, respectively. Variance in access to mating partners (component V) was a strong determinant of variance in $RS_m$

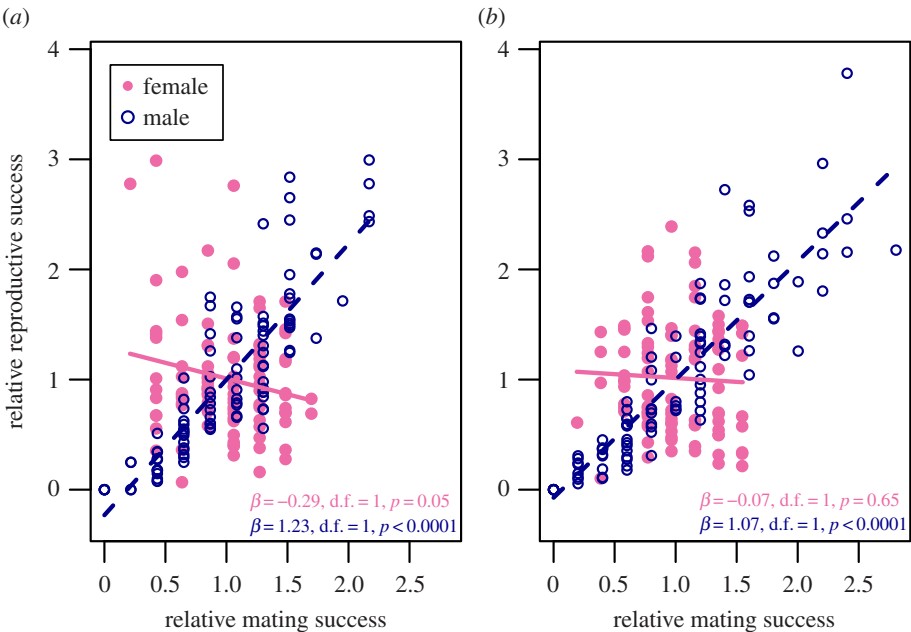

**Figure 1.** Sex-specific Bateman gradients in *M. annua* grown in two common gardens at (*a*) low density and (*b*) high density. (Online version in colour.)

**Table 1.** Opportunity for selection (*I*) and opportunity for sexual selection (*I_s*) in males and females in the low-density and high-density gardens. Opportunity for selection and opportunity for sexual selection were standardized by dividing by the square mean reproductive success or mean mating success. The opportunity for selection in males was corrected for binomial sampling errors in the measurement of paternity shares in each female (uncorrected values are indicated in parentheses). The 95% confidence intervals calculated on the basis of bootstrap replicates are provided in brackets.

| | low density | | high density | |
|---|---|---|---|---|
| | **female** | **male** | **female** | **male** |
| *I* | 0.28 [0.18 – 0.37] | 0.53 (0.53) [0.40 – 0.69] | 0.25 [0.19 – 0.31] | 0.63 (0.63) [0.45 – 0.81] |
| *I_S* | 0.12 [0.09 – 0.16] | 0.26 [0.19 – 0.35] | 0.11 [0.08 – 0.14] | 0.43 [0.32 – 0.56] |

in both gardens (figure 2); it was the largest variance component in 92% and 100% of the bootstrap replicates in the low- or high-density gardens, respectively. Securing paternity share at the extra-pair scale (VI) was a strong determinant of variance in RS_m in both gardens (figure 2). In both gardens, but to a greater extent in the low-density garden, some variance also emerged at the intra-pair scale, and this was not only because female seed production varied among pots (III), but also because some males failed to pollinate their associated female (I) and because their paternity share was variable when they did so (II). The significantly positive covariance between intra-pair and extra-pair reproductive success suggested that males that gained high reproductive output at the intra-pair scale also did so at the extra-pair scale in both gardens (figure 2, XI). Still, in both gardens, males with greater extra-pair mating success also sired a larger proportion of ovules on their extra-pair partners, as revealed by significant positive covariance between mating success and paternity share at the extra-pair scale (figure 2, IX).

## (c) Increased pollen dispersal distance allowed males to gain more mates

We found that males dispersing their pollen further acquired more mates in both gardens (table 2 and figure 3*a*). Pollen weight was not related to the proxy for male mating success

in either garden (table 2 and figure 3*b*). Pollen weight and dispersal distance were correlated in the high-density garden ($\Gamma = 0.35$, $t = 3.65$, d.f. = 93, $p = 0.0004$), but not in the low-density garden ($\Gamma = 0.16$, $t = 1.64$, d.f. = 98, $p = 0.11$).

In both gardens, increased pollen dispersal distance was positively associated with and explained a large proportion of extra-pair mating success (table 2 and figure 2). Increased pollen dispersal contributed to the positive associations found between intra-pair and extra-pair reproductive success, while pollen weight tended to decrease this association in the low-density garden (figure 2). In the low-density garden, increased pollen dispersal distance and pollen weight allowed males to sire larger proportion of ovules on extra-pair or intra-pair mates, respectively (table 2), but the explanatory power of the latter regression was low (figure 2). In the high-density garden, increased pollen dispersal was associated with larger mating success at both scales, and with larger paternity share on intra-pair females (table 2).

## (d) Simulations revealed opposite effects of plant density on Bateman metrics, depending on pollen dispersal abilities

With a simulated fixed ability to disperse pollen, we found that both *I* and *I_s* were larger at low than at high density when plants were distributed randomly over space (electronic

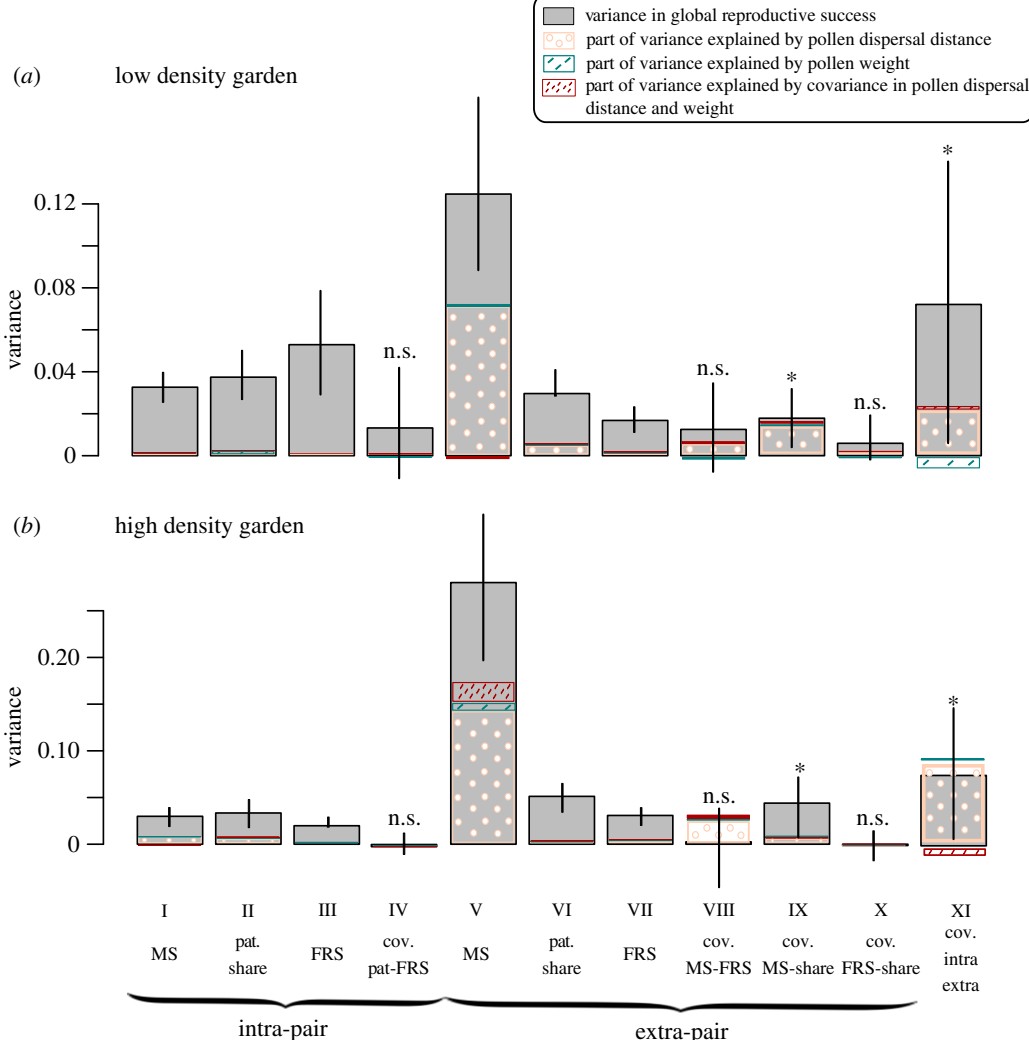

**Figure 2.** Decomposition of the variance in RS$_m$ at (a) low- and (b) high-density and explanatory power of pollen weight, dispersal distance and their covariance. I: variance in intra-pair mating success; II: variance due to the different reproductive outputs of the intra-paired female; III: variance due to paternity share on intra-pair female partners; IV: covariance between II and III; V: variance in extra-pair mating success; VI: variance due to the differences in reproductive success of extra-pair females; VII: variance due to differences in paternity share of extra-paired females; VIII: covariance between V and VI; IX: covariance between VI and VII; X: covariance between V and VII; XI: covariance between reproductive success at the intra-pair and extra-pair scale. Abbreviations: MS: mating success; Pat: paternity; FRS: reproductive success of the female partners; cov: covariance. Confidence intervals (95%) were calculated on the basis of bootstrap re-sampling of males. Significance of covariance terms was evaluated by computing the p-values corresponding to a null covariance term in the bootstrap distribution (*$p < 0.05$). (Online version in colour.)

supplementary material, figure S2d,e) but not when simulating a regular grid (electronic supplementary material, figure S2a,b). In neither of the simulated spatial conformations were Bateman gradients ($\beta_{SS}$) affected by density in the case of a fixed ability of males to disperse their pollen (electronic supplementary material, figure S2c,f). When simulating among-male variation in pollen dispersal distance, the impact of density on Bateman metrics depended on the distance of pollen dispersal relative to mean inter-individual distances: (1) with simulated long pollen dispersal distances, all three sexual selection metrics were larger at high compared to low density (figure 4 and electronic supplementary material, figure S2); (2) with simulated short pollen dispersal, Bateman metrics were larger at low compared to high density (figure 4 and electronic supplementary material, figure S2).

## 4. Discussion

Our study used classical Bateman statistics to quantify sexual selection in a flowering plant [5]. Variances in reproductive

and mating success proxies were larger in males of *M. annua* than in females, confirming that both natural and sexual selection had a greater opportunity to operate on males than on females, as is common in animals [5,7]. Previous work has obtained contrasted results on this point; variance in plant mating success was larger in males than in females in *Chamaelirium luteum* [46], while the opposite was true for other studies [47–49]. In our study, in addition to a difference in variance between the sexes, *M. annua* also conformed to the third Bateman principle: in both gardens, only males (i.e. not females) gained fitness benefits from having many mates. Males gained mates particularly through pollen dispersal over larger distances, rather than through pollen production, a result that points to sexual rather than fecundity selection. These results complement the body of work suggesting male–male competition as a selective force acting on several reproductive and vegetative plant traits [15,17–24]. Sexual selection should therefore act primarily on architectural traits that facilitate pollen dispersal in wind-pollinated plants [21,33] and, in insect-pollinated plants, on floral traits attracting pollinators that travel further

**Table 2.** Effect of pollen dispersal distance and pollen weight on several components of male reproductive success at the scale of intra-pair, extra-pair matings and on relative mating success at the global scale. Both pollen dispersal distances and pollen weight were standardized and analysed in bivariate models. Mean and s.d. are provided for each component of reproductive success.

| | intra-pair | | extra-pair | | | global | |
| | mating success | paternity share | partner reproductive success | mating success | paternity share | partner reproductive success | relative mating success |
|---|---|---|---|---|---|---|---|
| **low density** | | | | | | | |
| mean (± s.d.) | 0.82(±0.39) | 0.38(±0.26) | 121.45(±64.08) | 3.8(±2.22) | 0.16(±0.03) | 113.09(±31.29) | |
| pollen dispersal | $\beta = 0.45, p = 0.11$ | $\beta = 0.12, p = 0.16$ | $\beta = 3.91, p = 0.61$ | $\beta = 1.11, p <$ **0.0001***** | $\beta = 0.008, p =$ **0.02*** | $\beta = 2.10, p = 0.52$ | $\beta = 0.25, p <$ **0.0001***** |
| pollen weight | $\beta = 0.13, p = 0.65$ | $\beta = 0.17, p =$ **0.03*** | $\beta = -0.68, p = 0.93$ | $\beta = -0.13, p = 0.69$ | $\beta = -0.002, p = 0.47$ | $\beta = 3.31, p = 0.31$ | $\beta = -0.02, p = 0.59$ |
| **high density** | | | | | | | |
| mean (± s.d.) | 0.64(±0.48) | 0.22(±0.23) | 93(±44.14) | 4.33(±3.02) | 0.17(±0.04) | 88.62(±25.46) | |
| pollen dispersal | $\beta = 1.23, p <$ **0.0001***** | $\beta = 0.81, p <$ **0.0001***** | $\beta = -0.004, p = 0.99$ | $\beta = 1.67, p <$ **0.0001***** | $\beta = 0.002, p = 0.72$ | $\beta = 3.35, p = 0.26$ | $\beta = 0.41, p <$ **0.0001***** |
| pollen weight | $\beta = -0.15, p = 0.60$ | $\beta = 0.01, p = 0.91$ | $\beta = -9.21, p =$ **0.09** | $\beta = 0.39, p =$ **0.10** | $\beta = 0.001, p = 0.86$ | $\beta = 0.67, p = 0.82$ | $\beta = 0.05, p = 0.37$ |

The significance of each component of the reproductive success of males was evaluated using likelihood ratio tests: $^{.}p < 0.10$, $^{*}p < 0.05$, $^{**}p < 0.01$, $^{***}p < 0.001$. Results with a $p$-value $< 0.10$ are highlighted in bold.

away or on traits that promote more effective pollen deposition on pollinators [25].

Importantly, Bateman gradient estimates might be subject to a widely discussed statistical bias that is inherent in genetic estimates of mating success [4,27]. In such analyses, mating events that result in no, or few, fertilized eggs are necessarily ignored, so that variance in mating success may be overestimated (i.e. many male mates may remain below the detection threshold). In the likely scenario in which the male paternity share is strongly asymmetrical, our approach would allow the identification of the most successful males despite the low number of seeds sampled. It is not easy to identify an artificially induced variance in male mating success, but our positive Bateman gradients are unlikely to emerge only from random variation in the representation of males in the genotyped seeds.

Several features of our results indicate that they do capture true variance in the ability of males to access mates and are not just the result of sampling error. First, we observed spatial effects in mating patterns. Specifically, (1) most males tended to sire a large proportion of seeds on their local female, resulting in variance in the intra-pair paternity share; and (2) some males sired several of the sampled seeds on extra-pair females, increasing variance in the extra-pair paternity share. Second, males with many mates also showed a larger paternity share than expected at random. Third, a strong spatial component emerged in mate acquisition, suggesting that males dispersing their pollen over greater distances sired more seeds than expected by chance. Patterns of correlation in paternity, similar to those presented here, have recently been taken as indicative of the extent of sexual selection in plants [50], but they should ideally be estimated on the basis of more seeds sampled per female. Both the spatial effects and the variation in paternity share revealed by our approach suggest that males did differ from one another in their pollen efficiency, despite high polyandry.

It is also possible that the positive effects of pollen dispersal on male mating are simply a consequence of wind-pollination dynamics. For a given male, spreading pollen over more mates should reduce local mate and resource competition [33,35]. Nevertheless, the benefits of dispersing pollen widely likely could come at the cost of diluting the concentration of pollen (and lowering paternity share) per female. However, we did not find these trade-offs; if anything, males that dispersed their pollen over greater distances tended to have a higher share in paternity on the local female (at high density) or on distant females (at low density). This pattern suggests that males whose pollen travels further also have correlated traits that increase their paternity success in spite of potential pollen dilution. While the amount of pollen produced explained a small amount of the variance in male reproductive success, traits involved in the competitive ability of pollen might be correlated with pollen dispersal. This is reminiscent of many studies in animals where males in good condition tend to perform well for several fitness components at the same time, overriding potential trade-offs [51].

Both a sex-specific cost of reproduction and pollen limitation might lead to sex differences in Bateman metrics, two factors whose importance we did not evaluate. Females probably often incur a larger cost of reproduction than males, but the reverse could be true for wind-pollinated herbs in which

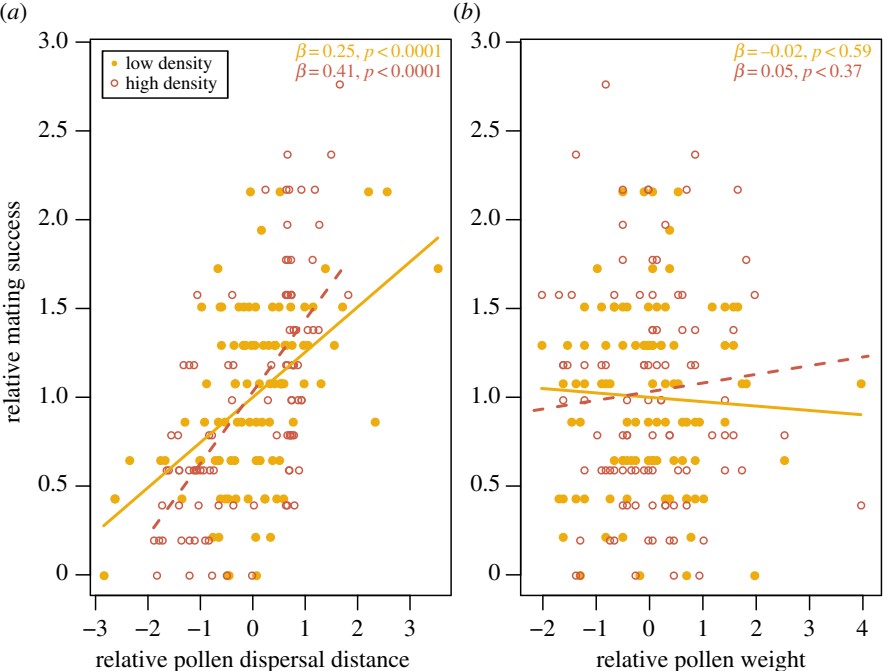

**Figure 3.** Relationship between mating success and (a) mean pollen dispersal distances and (b) pollen weight in *M. annua* grown at low density and at high density. (Online version in colour.)

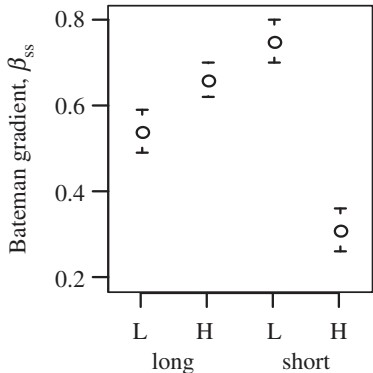

**Figure 4.** Effect of simulated low and high density (L versus H) on the Bateman gradient when male abilities to disperse pollen are variable. We implemented either a long or short mean dispersal distance of pollen compared to typical distances between males and females (long and short, respectively). Dots represent mean values, and error bars correspond to one standard deviation.

males produce large amounts of pollen [52]. If so, the larger opportunity for selection reported here for males might in part reflect among-male differences in a capacity to harvest resources. In species with a larger female reproductive cost, among-female variation in resource acquisition might dramatically increase variance in female reproductive success, which has commonly been found in plants [15,47–49], regardless of sexual selection. In our experiment, female reproductive success was independent of access to mates, but positive Bateman gradients in females are nevertheless expected under pollen-limited conditions [53]. Pollen limitation is unlikely to have been important in our experiment, in which females were close to a male in both gardens, and is probably rarely important in natural populations of *M. annua*, which tend to be dense [54]. At low density, pollen limitation might, however, be important in many species, as for sperm limitation in broadcast spawners, where variance

in female reproductive success is typically larger at lower densities [31].

Our simulations similarly suggested that the intensity of sexual selection in plants may be density-dependent, albeit constrained by the scale at which pollen is dispersed relative to the spatial distribution of potential mates. Increased variance in both reproductive and mating success was predicted with increasing distance between the sexes, because the skewness of pollen dispersal kernels enhanced differences in the ability of males to disperse pollen successfully at a lower density. In a randomly arranged population, some males will by chance experience a more female-biased neighbourhood than others, as can happen in natural populations [55], and might thus enjoy both a higher mating and a higher reproductive success. Such a stochastic effect of decreased density was cancelled in populations with a uniform distribution of plants, i.e. a regular grid in our experiment.

While plants may have little genetic control on their relative positions, our simulations also indicated that sexual selection may have non-neutral density-dependent effects on traits involved in pollen export. The opportunity for sexual selection increased at lower densities only when a few males dispersed their pollen further than average males, thus obtaining disproportionate fitness gains by mating with more mates (when most males dispersed their pollen over shorter distances than the typical inter-individual distance). By contrast, the opportunity for sexual selection increased at higher densities only when a few males dispersed more pollen in their immediate vicinity than average males. In this case, local dispersal should be disproportionately favoured by concentrating pollen on the closest females where they can outcompete other pollen donors. Sexual selection might bring about the evolution of strategies (or plastic responses to variation in plant density) that allow males to disperse most of their pollen either locally or far away, depending on the spatial distribution of their prospective mates. These simulation results echo findings in

broadcast spawners, where sperm traits that increased competitive performance were favoured by selection at high density, whereas sperm traits facilitating the localization of rare eggs were favoured at low density [31,34].

By applying a mate-centred approach, and by decomposing male reproductive success into different components, our study suggests that a capacity for enhanced pollen dispersal is associated with larger success in accessing mates, which in turn is the main determinant of male fitness—a result that might not always hold. Both our experimental results and our simulations revealed that the spatial conformation of a population may significantly affect the strength and direction of sexual selection. Bateman metrics and variance decomposition, initially developed to quantify how sexual selection operates in animals, thus have the potential to capture this variation and to inform us on selection on traits that affect the spatial dispersal of pollen.

Data accessibility. Our manuscript uses already published and public data.

Authors' contributions. J.T. performed the original experiment, J.T. and P.D. performed the statistical analysis, J.T. undertook the computer simulations, J.T. drafted the manuscript and all authors contributed to the final version of the manuscript.

Conflict of interest. We declare we have no competing interests.

Funding. J.T. was supported by a grant to J.R.P. by the Swiss National Science Foundation.

Acknowledgements. We thank Tim Janicke and Etienne Klein for useful discussions. The experiment was undertaken 'Plateforme des Terrains d'Expériences du LabEx CeMEB' (Montpellier, France).

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
