## [Reviewer comments · Proceedings of the Royal Society B: Biological Sciences]

Review History

RSPB-2019-0532.R0 (Original submission)

Review form: Reviewer 1

Recommendation

Accept with minor revision (please list in comments)

Scientific importance: Is the manuscript an original and important contribution to its field?

Excellent

General interest: Is the paper of sufficient general interest?

Excellent

Quality of the paper: Is the overall quality of the paper suitable?

Acceptable

Is the length of the paper justified?

Yes

Should the paper be seen by a specialist statistical reviewer?

No

Do you have any concerns about statistical analyses in this paper? If so, please specify them explicitly in your report.

No

It is a condition of publication that authors make their supporting data, code and materials available - either as supplementary material or hosted in an external repository. Please rate, if applicable, the supporting data on the following criteria.

Is it accessible?

No

Is it clear?

No

Is it adequate?

No

Do you have any ethical concerns with this paper?

No

Comments to the Author

This manuscript reports on a study of sexual selection in plants using a dioecious, wind-pollinated herb. It shows that, like in animals, that male fitness is more variable than female fitness, with male fitness increasing with increasing access to mates, whereas this is not the case for female fitness. As such, it confirms earlier studies in animals and some studies in plants. Where it differs from previous plant studies is that it takes a Bateman-gradient approach. That said, earlier work on the topic needs to be acknowledged and discussed. The authors are building on a foundation and by not citing this foundational work they are preventing naïve readers from knowledge of a richness of studies approaching the question of what limits male vs. female reproductive success. I am thinking of studies such as Queller 1983 Nature, Meagher 1986 Am Nat, Schoen 1986 Evolution, Meagher 1991 Am Nat, to name a few. The work presented in the manuscript is well executed, important, fairly well explained, and makes good use of the study system.

Abstract

Line 9 - the authors use three related phrases: reproductive success, mating success, and mating effort. I don't understand why the last of these is used, and feel that the authors need to define what they mean by reproductive success vs. mating success early on - perhaps in line 20.

Line 10 - should read 'associated with males' abilities...'

Line 12-13 - this is an odd way to end the Abstract. Is this really what the authors want the readers to take away? If so, I do not see that they have sufficiently explained this in the ms.

Introduction

Line 30 - get rid of mating effort, which is not explained.

Line 42-43 - this attempt to parse things in a way that allows the authors to say they are the first comes across as unnecessary.

Lines 49-68 - I don't understand what the authors are trying to get at here. What do they mean by 'limiting case' and why do they think that anyone would assume equal mating success? This dichotomy seems like a straw man that doesn't help the reader. 'Notorious' (defn: famous or well known, typically for some bad quality or deed) does not seem like a word that should be applied

to plants.

Lines 89-98 – this paragraph seems misplaced. It does not ‘introduce’ something about the study in a useful way to the reader.

Lines 112-117 – I advise the authors to lose the terms ‘budget’ and ‘direct.’ They simply add an unnecessary layer to the framework. I am not alone in thinking that fecundity selection and sexual selection are not the same thing and should be differentiated from each other (Arnold, Shine, Losos, Blanckenhorn). This paragraph muddles the two.

Discussion

The Discussion starts well, but again, acknowledging previous work by other authors belongs here.

Line 418 – consider citing De Cauwer et al. 2010 Mole Ecol.

Lines 422-432 – unless male plants can judge density (how would that happen? I can see females doing via pollen deposition), this paragraph does not make sense to me.

Lines 443-445 – I appreciate the analogy to broadcast spawners.

The finding that pollen dispersal, but not their measure of the number of pollen grains (pollen weight), affects mating success says to me that this really is sexual selection at work and not merely fecundity selection. The authors might consider adding something to this effect.

Review form: Reviewer 2

Recommendation

Major revision is needed (please make suggestions in comments)

Scientific importance: Is the manuscript an original and important contribution to its field?

Excellent

General interest: Is the paper of sufficient general interest?

Excellent

Quality of the paper: Is the overall quality of the paper suitable?

Excellent

Is the length of the paper justified?

Yes

Should the paper be seen by a specialist statistical reviewer?

No

Do you have any concerns about statistical analyses in this paper? If so, please specify them explicitly in your report.

No

It is a condition of publication that authors make their supporting data, code and materials available - either as supplementary material or hosted in an external repository. Please rate, if applicable, the supporting data on the following criteria.

Is it accessible?

Yes

Is it clear?

Yes

Is it adequate?

N/A

Do you have any ethical concerns with this paper?

No

Comments to the Author

I have reviewed the manuscript entitled “Do metrics of sexual selection conform to Bateman’s principles in a wind-pollinated plant?” for publication in Proc B. This MS reports a study in which 100 male plants and 100 female plants were planted in gardens of either low (n=1) and high density (n=1). After a four weeks of mating, all adults and seeds (about 600+ per garden) were genotyped to infer paternity. Authors used these data to estimate metrics of sexual selection, which are widely used in animal biology but have apparently not been applied to plant systems so far. Interestingly, the data suggest that plants show similar patterns of sexual selection than what is generally observed in animals. The authors further decompose the variance observed in male reproductive success along multiple fitness components, contrasting male fitness acquired through a close female plant (planted in the same pot) and the one gained through all the other females in the population. Most of the variance in fitness seems to arise from the mating success gained with the other females in the population. Finally, the authors ran computer simulations to explore the role of plant density on the operation of sexual selection, revealing that weather sexual selection is stronger at low or high density ultimately depends on males’ ability to disperse.

In general, I enjoyed reading the MS. I am not an expert in plant biology and, given how widespread Bateman metrics are in animal biology, it strikes me why such a study had not been done before. The MS reads very well, the results are interesting, and the MS leads to interesting follow-up questions about the relationship between sexual selection and plant density. I have a few comments that are mainly minor and meant to be useful.

General comments

(1) I appreciate that authors clearly say that the experimental design was not done to compare density because low and high density were not replicated (line 164-166). One cannot compare two treatments that have not been replicated. The authors have successfully refrained themselves from comparing these two treatments, with the exception of the paragraph line 327-332, which should be deleted (along with any other potential comparison I may have missed elsewhere).

(2) The variance decomposition approach is well-suited for such an experimental design with a paired female and extra-paired females. I however feel that more details are required. First, authors say that all covariances were estimated (line 238), but the results show only 5 covariances (Fig. 2). Please clarify why e.g. cov MS-PS were not explored in the intra-pair fitness route? Is it that some cannot be estimated, or are not biologically interesting? Is it that covariances can be estimated only within each route (intra vs inter)? I feel that more details should be given here, and please use consistent abbreviation, e.g. does Pat share = Pat = share? Second, my understanding of variance decomposition approach is that confidence intervals can be estimated through bootstrapping, which would be handy to formally estimate how good each variance value is known and to compare variance components to one another (e.g., Fig 2 in Janicke et al, Am Nat, 2015). I believe this is a necessary addition, as it would allow authors to tell if variances are statistically different from one another which, strictly speaking, cannot be done in the current version of the MS.

Specific comments

156: What is the rationale for raising all plants at low density? Is it possible that the density of young plants will influence adult sexual traits, such as e.g. pollen dispersal traits?

179-180: Given that relatively few seeds have been genotyped per female (about 6.5 per female plant), it is likely that sires have been missed in the paternity analyses. How is it supposed to affect the results?

334-344: the simulation is a nice part of the MS and, in my opinion, it would be nice to have a figure showing these results.

Table 1: Does the correction for Binomial sampling error not change the results, or is it a typo?

Table 2: Are all betas standardised? Some are very large and non-significant (e.g., extra-pair partner RS), while others are tiny and significant (extra-pair PS).

Decision letter (RSPB-2019-0532.R0)

17-Apr-2019

Dear Dr Tonnabel:

Your manuscript has now been peer reviewed and the reviews have been assessed by an Editor. The reviewers' comments (not including confidential comments to the Editor) and the comments from the Associate Editor are included at the end of this email for your reference. As you will see, the reviewers and the Editors have raised some concerns with your manuscript and we would like to invite you to revise your manuscript to address them.

Research ethics:

Use of animals and field studies:

Please submit a copy of your revised paper within three weeks. If we do not hear from you within this time your manuscript will be rejected. If you are unable to meet this deadline please let us know as soon as possible, as we may be able to grant a short extension.

Best wishes,
Proceedings B
<mailto:proceedingsb@royalsociety.org>

Associate Editor

Comments to Author:

I have received two reviews of this manuscript, and both reviewers have attested to the novelty and importance of this work. I am also of the opinion that it will make a very noteworthy contribution to sexual selection theory in plants. While its main finding (that male fitness is more variable than female fitness) echoes many similar studies done on animals, this is the first plant study to use a Batemanian-gradient approach to compare variation in mating success. For some time there has been debate about whether mating patterns are likely to be similar in plants and animals and this manuscript helps to settle the debate.

Perhaps the most serious criticism from reviewers (Rev 1) was that the authors have not engaged properly with some of the foundational plant literature that precedes this work. This is an important omission which the authors will need to fix before the publication of this manuscript.

Reviewer 1 also pointed out the use of three related terms (reproductive success, mating success, and mating effort) which can be quite confusing to readers who are unfamiliar with this subject material. Indeed, I also found the use of this terminology a little difficult, as the explanations only came much later. I think it is very important to define these terms early in the manuscript and also to make sure there is no duplication of terminology.

Reviewer 2 makes a good point about comparative statistics and that in the present manuscript, it is impossible to properly compare variance components. The reviewer suggests a bootstrapping approach to solve this problem. I think that this would make a good addition to the present manuscript.

While one reviewer suggests major revision and the other suggests minor revision, I found that the introduction was poorly structured and often very hard to follow because explanations often tended to come too late. In places, the sentence structure was a little hard to follow and I urge the authors to read the manuscript slowly, out loud to help identify areas with somewhat complicated wording. In addition to the points made by the reviewers, I also have a number of points and suggestions which I hope will improve the reading of this manuscript. Please forgive me if any points made by the reviewers are repeated.

L3: The term anisogamous may not be familiar to everyone. It needs an associated definition for first time use. Abstract is perhaps not the best place to start using it. Just say gametes of different size

L5-7: sentence too long and clumsy

Introduction: the terms reproductive success and mating success are used within the first few lines of the introduction. The terms are quite similar and to many the differences may be subtle or even difficult to tease apart. But understanding their difference is crucial to this manuscript. Authors need to explain the difference right from the start of the ms. Later in the ms, the authors state that genetic markers are used to calculate reproductive success and this makes for even more confusion because obviously genetic markers only pick up mating events that result in progeny, which starts to blur the lines between mating success and reproductive success. Right from the start, the authors need to say how each of these metrics should be calculated and how they are typically calculated in reality.

L16 and ?

L36: Explain why Bateman's principle may be different for modular organisms before you pose this question. The reasoning behind this question only comes much later. So at this point there is no logical lead-in to this question. It's a little like starting a joke with the punch-line.

L54: It may help the reader to understand the concept of mating success better if you actually give an example of how it is calculated

L57: No idea what this limiting case is about

L57-59: This feels like a straw man as it is almost impossible for all individuals in a population to receive pollen from all donors (unless the population is very small and discrete).

L72: given that mating success should be counts of mating events, it does not really make sense that you could calculate it using molecular tools. This creates confusion about what you actually mean by mating success. You need to make sure that this issue is addressed much earlier in the manuscript (I see that it is addressed later). I also think that you cannot call this mating success. It is a surrogate for mating success. I suggest that throughout the ms, you call it what it is and motivate that it is in fact a reasonable surrogate for mating success.

L93-95: This was the purpose of our recent review in AoB 2019, Minnaar and Anderson, Plant-pollinator interactions along the pathway to paternity. Despite describing the male fitness pathway through the lens of animal pollinated plants, it has many intersections with your own paper in discussions of sexual selection, bateman's predictions, analogies to broadcast spawning, and identifying the problems associated with using genetic tools for calculating mating success.

L95: Never good to lead with such a perspective as it forces the reader to go back and find the perspective you are referring to. Just state the perspective

L101: Not sure what you mean by "dam." The word is used at least twice in this para

L128, 129: these terms are not properly explained here and because the reader does not know exactly what you mean, it makes the rest hard to follow. I saw that they were explained (in terms of calculations) in the methods. Perhaps it should be done here.

L134: I was very surprised to see any hypotheses here. This work is completely hypothesis-driven and I would like to see explicit hypotheses developed here.

L152: It was never made very clear why this experiment is done at 2 different densities. This needs to be clear and it should be worked into the hypotheses what you expect to find by this and why.

L153: It is also not very clear at this stage why the paired design was important. I think it needs to be motivated more clearly.

L317: what is meant by pollen distance?

L334, 335: What is meant by non-trivial? It feels meaningless and non-quantitative

L451: spatial conformation - Do you mean density?

Reviewer(s)' Comments to Author:

Referee: 1

Comments to the Author(s)

This manuscript reports on a study of sexual selection in plants using a dioecious, wind-pollinated herb. It shows that, like in animals, that male fitness is more variable than female

fitness, with male fitness increasing with increasing access to mates, whereas this is not the case for female fitness. As such, it confirms earlier studies in animals and some studies in plants. Where it differs from previous plant studies is that it takes a Bateman-gradient approach. That said, earlier work on the topic needs to be acknowledged and discussed. The authors are building on a foundation and by not citing this foundational work they are preventing naïve readers from knowledge of a richness of studies approaching the question of what limits male vs. female reproductive success. I am thinking of studies such as Queller 1983 *Nature*, Meagher 1986 *Am Nat*, Schoen 1986 *Evolution*, Meagher 1991 *Am Nat*, to name a few. The work presented in the manuscript is well executed, important, fairly well explained, and makes good use of the study system.

Abstract

Line 9 - the authors use three related phrases: reproductive success, mating success, and mating effort. I don't understand why the last of these is used, and feel that the authors need to define what they mean by reproductive success vs. mating success early on - perhaps in line 20.

Line 10 - should read 'associated with males' abilities...'

Line 12-13 - this is an odd way to end the Abstract. Is this really what the authors want the readers to take away? If so, I do not see that they have sufficiently explained this in the ms.

Introduction

Line 30 - get rid of mating effort, which is not explained.

Line 42-43 - this attempt to parse things in a way that allows the authors to say they are the first comes across as unnecessary.

Lines 49-68 - I don't understand what the authors are trying to get at here. What do they mean by 'limiting case' and why do they think that anyone would assume equal mating success? This dichotomy seems like a straw man that doesn't help the reader. 'Notorious' (defn: famous or well known, typically for some bad quality or deed) does not seem like a word that should be applied to plants.

Lines 89-98 - this paragraph seems misplaced. It does not 'introduce' something about the study in a useful way to the reader.

Lines 112-117 - I advise the authors to lose the terms 'budget' and 'direct.' They simply add an unnecessary layer to the framework. I am not alone in thinking that fecundity selection and sexual selection are not the same thing and should be differentiated from each other (Arnold, Shine, Losos, Blanckenhorn). This paragraph muddles the two.

Discussion

The Discussion starts well, but again, acknowledging previous work by other authors belongs here.

Line 418 - consider citing De Cauwer et al. 2010 *Mole Ecol*.

Lines 422-432 - unless male plants can judge density (how would that happen? I can see females doing via pollen deposition), this paragraph does not make sense to me.

Lines 443-445 - I appreciate the analogy to broadcast spawners.

The finding that pollen dispersal, but not their measure of the number of pollen grains (pollen weight), affects mating success says to me that this really is sexual selection at work and not merely fecundity selection. The authors might consider adding something to this effect.

Referee: 2

Comments to the Author(s)

I have reviewed the manuscript entitled "Do metrics of sexual selection conform to Bateman's principles in a wind-pollinated plant?" for publication in *Proc B*. This MS reports a study in which 100 male plants and 100 female plants were planted in gardens of either low ($n=1$) and

high density ($n=1$). After a four weeks of mating, all adults and seeds (about 600+ per garden) were genotyped to infer paternity. Authors used these data to estimate metrics of sexual selection, which are widely used in animal biology but have apparently not been applied to plant systems so far. Interestingly, the data suggest that plants show similar patterns of sexual selection than what is generally observed in animals. The authors further decompose the variance observed in male reproductive success along multiple fitness components, contrasting male fitness acquired through a close female plant (planted in the same pot) and the one gained through all the other females in the population. Most of the variance in fitness seems to arise from the mating success gained with the other females in the population. Finally, the authors ran computer simulations to explore the role of plant density on the operation of sexual selection, revealing that whether sexual selection is stronger at low or high density ultimately depends on males' ability to disperse.

In general, I enjoyed reading the MS. I am not an expert in plant biology and, given how widespread Bateman metrics are in animal biology, it strikes me why such a study had not been done before. The MS reads very well, the results are interesting, and the MS leads to interesting follow-up questions about the relationship between sexual selection and plant density. I have a few comments that are mainly minor and meant to be useful.

General comments

(1) I appreciate that authors clearly say that the experimental design was not done to compare density because low and high density were not replicated (line 164-166). One cannot compare two treatments that have not been replicated. The authors have successfully refrained themselves from comparing these two treatments, with the exception of the paragraph line 327-332, which should be deleted (along with any other potential comparison I may have missed elsewhere).

(2) The variance decomposition approach is well-suited for such an experimental design with a paired female and extra-paired females. I however feel that more details are required. First, authors say that all covariances were estimated (line 238), but the results show only 5 covariances (Fig. 2). Please clarify why e.g. cov MS-PS were not explored in the intra-pair fitness route? Is it that some cannot be estimated, or are not biologically interesting? Is it that covariances can be estimated only within each route (intra vs inter)? I feel that more details should be given here, and please use consistent abbreviation, e.g. does Pat share = Pat = share? Second, my understanding of variance decomposition approach is that confidence intervals can be estimated through bootstrapping, which would be handy to formally estimate how good each variance value is known and to compare variance components to one another (e.g., Fig 2 in Janicke et al, Am Nat, 2015). I believe this is a necessary addition, as it would allow authors to tell if variances are statistically different from one another which, strictly speaking, cannot be done in the current version of the MS.

Specific comments

156: What is the rationale for raising all plants at low density? Is it possible that the density of young plants will influence adult sexual traits, such as e.g. pollen dispersal traits?

179-180: Given that relatively few seeds have been genotyped per female (about 6.5 per female plant), it is likely that sires have been missed in the paternity analyses. How is it supposed to affect the results?

334-344: the simulation is a nice part of the MS and, in my opinion, it would be nice to have a figure showing these results.

Table 1: Does the correction for Binomial sampling error not change the results, or is it a typo?

Table 2: Are all betas standardised? Some are very large and non-significant (e.g., extra-pair partner RS), while others are tiny and significant (extra-pair PS).

Author's Response to Decision Letter for (RSPB-2019-0532.R0)

See Appendix A.

Decision letter (RSPB-2019-0532.R1)

08-May-2019

Dear Dr Tonnabel

I am pleased to inform you that your manuscript RSPB-2019-0532.R1 entitled "Do metrics of sexual selection conform to Bateman's principles in a wind-pollinated plant ?" has been accepted for publication in Proceedings B.

The referee(s) have recommended publication, but also suggest some minor revisions to your manuscript. Therefore, I invite you to respond to the referee(s)' comments and revise your manuscript. Because the schedule for publication is very tight, it is a condition of publication that you submit the revised version of your manuscript within 7 days. If you do not think you will be able to meet this date please let us know.

Sincerely,

Proceedings B
<mailto:proceedingsb@royalsociety.org>

Associate Editor:
 Board Member
 Comments to Author:

I think you have done an excellent job of attending to the comments of the two reviewers and I feel that the changes have made for a much clearer manuscript. I have a few minor suggestions

(mostly regarding sentence structure) which I hope will further improve the manuscript (see below).

L23: males' - male

L25: females' - female

L25: You say that female success is about the production of viable ovules, but I was thinking that it may be more than just production of ovules: Seed production (i.e. after the ovule is fertilized) is also very resource-intensive for the female pathway and I can imagine that fruit abortion happens when resources are scarce. I may be wrong, but perhaps you need to include resource allocation to seed production here: In contrast, female reproductive success should depend on their ability to allocate resources to the production of viable ovules and seeds rather than the probability of having ovules fertilized.

L27: delete comma after and

L60, 61: ...potentially resulting in extremely high...

L78: a plant's

L78: split sentence: ...plant level. However, it is also...

L80: profitable-useful

L80-84: Split sentence:....distance. Consequently plant size and architecture may be selected for either through....

L85: larger pools of resources that can be allocated to gamete production;...

293: remove second "at"

L349: ...[46], while the opposite was true for other studies [47-49].

L385: I am not fond of how you start this paragraph. First, I think you need to connect better to the previous para: While our results suggest X and Y, it is also possible that the positive effects of pollen dispersal on male mating are simply a consequence of wind-pollination dynamics.

Then you go into explaining what one would find if it is just a wind dispersal consequence but this is not clear until later. Try writing: If this is the case, then we would expect A and B. Then you can go into what you found: However we did not find these trade-offs.

Decision letter (RSPB-2019-0532.R2)

31-May-2019

Dear Dr Tonnabel

I am pleased to inform you that your manuscript entitled "Do metrics of sexual selection conform to Bateman's principles in a wind-pollinated plant ?" has been accepted for publication in Proceedings B.

Open Access

Paper charges

Sincerely,

Appendix A

Dear Editor,

Please find attached a revised version of our manuscript “*Do metrics of sexual selection conform to bateman’s principles in a wind-pollinated plant ?*” by Jeanne Tonnabel, Patrice David, John R. Pannell which we wish to resubmit for publication in *Proceedings of the Royal Society B*.

This manuscript has been evaluated by Associate Editor (AE) Prof. Bruce Anderson, and two anonymous reviewers. They converged on their conclusions that our study and its discussion advance the field of plant sexual selection. By adopting a Batemanian approach and discussing its scope and limits, our study may pave the way for future study of this kind in plants. We believe that applying the Bateman framework in plant clades will likely help understand how sexual selection operates in plants. Despite these merits, both reviewer and AE identified three main concerns that required a revision: (i) a more thorough review of foundational work performed on plant sexual selection, (ii) the need for a better structured Introduction including the definition of key concepts and key terms used in the manuscript before discussing them and (iii) the need to estimate uncertainty in variance components by adding a bootstrap analysis. Below, we first describe the main changes that we performed before answering in more details to the AE and reviewer’s comments. We thank both reviewers and AE for their suggestions on our manuscript. We believe that these changes have widened the scope of our manuscript and strengthened its conclusions. We hope that you find this new version acceptable for publication in *Proceedings of the Royal Society B*.

(i) Better highlight how our study relates to foundational literature on plant reproduction thanks to a more thoroughly literature review : both reviewer 1 and AE identified that our manuscript was lacking a precise review on previous work on sexual selection. We agree that it is fundamental that we highlight how our work relates to previous work on plant reproduction and sexual selection. We have now referred to a much wider list of references on these aspects (as shown in the list of added references at the bottom of this response letter). In particular, we now refer in the Introduction section to a list of key studies that suggested that male-male competition could trigger changes in vegetative and reproductive traits (see lines **-*). We also now refer to the AE’s review in AoB which introduces the idea of competitive interactions between pollen grains on the pollinators body (lines **-*). In the discussion, we again refer now to studies that investigated the effect of male-male competition on several plant traits and to older studies that estimated variance in reproductive success through the male and female function which seemed extremely relevant in the context of our results. Because of space constraint and because we believe that it is important to devote space to discuss the scope and limit of the Bateman approach in plants, we however, cannot be exhaustive in the literature review on plant sexual selection. All references pointed out by both reviewer 1 and AE are now cited in the manuscript. We hope that this revised version will better put forward how our study is related to previous work.

(ii) Clarification of mating and reproductive success and providing a better structure in the Introduction section : as pointed out both by reviewer 1 and AE, the previous version of the Introduction was hard to follow, mainly, because we were referring to key concepts without defining them properly or defining them too long after their first use. The definition of mating and reproductive success, central to Bateman gradients calculation, in particular came too late in the Introduction. We have reviewed the complete structure of the Introduction section by carefully checking that the flow of information now comes in a logical way (please note that the paragraphs in the track-changed version are not highlighted in blue if they simply have been moved). We now provide definitions of mating and reproductive success in the very first paragraph of the Introduction section. This paragraph is followed by a paragraph describing how mating and reproductive success are assessed and why estimating mating success through paternity analysis may be problematic. In order to clarify this last point, we have also termed our mating success estimate ‘genetically-based mating success proxy’ to be clearer on the metric that we used and kept that definition constant throughout the manuscript.

Finally, some confusion was caused by the fact that we sometimes interchanged ‘mating success’ with ‘mating effort’, which should rather refer to the proportion of resources allocated to mate acquisition than to mate numbers. We have now avoided the use of ‘mating effort’ throughout the text.

(iii) Addition of uncertainty estimations on variance partitioning using a bootstrap analysis: we followed reviewer 2’s advice to estimate uncertainty in the variance of each male reproductive success components. Following Janicke *et al.* 2015 we computed 10,000 bootstrap replicates by randomly sampling males (and their associated mating variables, mating success, paternity share and fecundity of partners, intra and extra) and recalculated variance components for each replicate. We have now reported in figure 2 the 95% confidence intervals for each component which allows to graphically examine the uncertainty associated with their estimations. We did not use these bootstraps to estimate differences between fitness components because it was not the aim of the manuscript. But, since we claimed that differences in extra-pair (genetically-based) mating success was the major determinant of male reproductive success, we have added the information about the percentage of bootstrap replicates for which this reproductive success component was highest (see lines **-*).

We also performed bootstraps to test the significance of covariances which was indeed a more robust test than non parametric tests used before (a few borderline results became nonsignificant, while no nonsignificant test became significant). In the new version of the manuscript, we therefore have less significant covariance parameters than previously. However this concerns covariances that were quite weak and therefore not discussed in the previous manuscript. The covariances that were discussed were large and stayed significant even with the more robust bootstrap approach. All these new methods are explained on lines **-*.

Finally, we also used the same bootstrap procedure to assess uncertainty in the estimation of variance in global mating and reproductive success (assessed genetically) in both males and females. These results are central to the manuscript and this bootstrap analysis confirmed that there was a significant difference between males and females in both mating and reproductive success as none of the 95% confidence intervals overlapped. These new methods are explained on lines **-* , the new results are reported in the Table 1 and described on lines **-*.

Janicke, T., David, P., Chapuis, E. (2015) Environment-dependent sexual selection: Bateman's parameters under varying levels of food availability. *The American Naturalist*, **185**: 756-768.

Associate Editor

I have received two reviews of this manuscript, and both reviewers have attested to the novelty and importance of this work. I am also of the opinion that it will make a very noteworthy contribution to sexual selection theory in plants. While its main finding (that male fitness is more variable than female fitness) echoes many similar studies done on animals, this is the first plant study to use a Batemanian-gradient approach to compare variation in mating success. For some time there has been debate about whether mating patterns are likely to be similar in plants and animals and this manuscript helps to settle the debate.

Perhaps the most serious criticism from reviewers (Rev 1) was that the authors have not engaged properly with some of the foundational plant literature that precedes this work. This is an important omission which the authors will need to fix before the publication of this manuscript.

Author response : As described above in our general letter, we now give an in-depth review of the literature on plant reproduction relevant to our work. We substantially modified both the Introduction and Discussion section to better highlight how our study relates to foundational literature on Plant reproduction (see lines **-* , **-* and **-* for some examples on this point). We have now added all the references pointed out both by the AE and reviewers, along with some others (please find the list of added references at the end of this response letter). We believe that these changes allow to better highlight the novelty of our work and its link with classical work on plant mating.

Reviewer 1 also pointed out the use of three related terms (reproductive success, mating success, and mating effort) which can be quite confusing to readers who are unfamiliar with this subject material. Indeed, I also found the use of this terminology a little difficult, as the explanations only came much later. I think it is very important to define these terms early in the manuscript and also to make sure there is no duplication of terminology.

Author response : These terms are indeed central in our manuscript as reproductive and mating success are two metrics allowing to estimate Bateman gradients and referring to the number of offspring and of mating partners respectively. Mating effort referred to the resource allocation to access to mates (e.g. traits associated with pollen export and dispersal in our case) relative to total resources. We agree with both reviewer 1 and AE that the previous version of our manuscript was lacking clear definition of several terms that may not be straightforward for the reader. The comments made by reviewer 1 also made us realize that we were often interchanging « mating success » with « mating effort » which created confusion. We have now substantially modified the Introduction section to make sure that all key terminology is defined early on (in the first and second paragraphs of the new introduction version) and used consistently throughout the text (see lines **-** and **-** for instance, see below). We have also now replaced all references to « mating effort » with « mating success ».

Reviewer 2 makes a good point about comparative statistics and that in the present manuscript, it is impossible to properly compare variance components. The reviewer suggests a bootstrapping approach to solve this problem. I think that this would make a good addition to the present manuscript.

Author response : We agree with reviewer 2 that adding a bootstrap analysis on the calculation of variance components is valuable and we have added such an analysis following the methodology pointed out by reviewer 2 (Janicke *et al.* 2015). As pointed out in our general response above, this new analysis allowed us to document the uncertainty in variance components estimations and to provide a better test for the significance of covariance terms.

While one reviewer suggests major revision and the other suggests minor revision, I found that the introduction was poorly structured and often very hard to follow because explanations often tended to come too late. In places, the sentence structure was a little hard to follow and I urge the authors to read the manuscript slowly, out loud to help identify areas with somewhat complicated wording. In addition to the points made by the reviewers, I also have a number of points and suggestions which I hope will improve the reading of this manuscript. Please forgive me if any points made by the reviewers are repeated.

Author response : We have now revised our Introduction section by clarifying the key terminology early on, by better reviewing the literature on plant reproduction and by providing more structure in the new Introduction section (see our general response above for a complete description). We have also thoroughly checked for complicated sentences and wording.

L3: The term anisogamous may not be familiar to everyone. It needs an associated definition for first time use. Abstract is perhaps not the best place to start using it. Just say gametes of different size

Author response : We now refer to gametes of different sizes in the abstract (line **) and define the term anisogamy when we first introduce this notion in the Introduction section (see line **).

L5-7: sentence too long and clumsy

Author response : This sentence has been changed (lines **-**).

Introduction: the terms reproductive success and mating success are used within the first few lines of the introduction. The terms are quite similar and to many the differences may be subtle or even difficult to tease apart. But understanding their difference is crucial to this manuscript. Authors need

to explain the difference right from the start of the ms. Later in the ms, the authors state that genetic markers are used to calculate reproductive success and this makes for even more confusion because obviously genetic markers only pick up mating events that result in progeny, which starts to blur the lines between mating success and reproductive success. Right from the start, the authors need to say how each of these metrics should be calculated and how they are typically calculated in reality.

Author response : We agree that it is crucial that the readers understand both the differences between reproductive and mating success and the challenges associated with the estimation of mating success in general and in plants in particular. To circumvent this issue, we have defined mating and reproductive success right from the beginning of the Introduction and placed a paragraph explaining how mating success is typically calculated and why it only captures a fraction of the mating events (see lines **-*), see also above and below for further explanation on that point).

L16 and ?

Author response : We have changed the leading sentence to « Darwin [1] himself introduced the notion of sexual selection and attributed sex-roles whereby males tend to compete with each other to access females, whereas females often chose their male mates. » (lines **-*) which better introduces the concept of sexual selection we are studying in the manuscript.

L36: Explain why Batemans principle may be different for modular organisms before you pose this question. The reasoning behind this question only comes much later. So at this point there is no logical lead-in to this question. It's a little like starting a joke with the punch-line.

Author response : We have now completely reorganized the Introduction section to make sure that the flow of information comes in the right order. In particular, we introduce the reasons why sexual selection and Bateman principles may apply differently to modular organisms compared to most animals before asking the question (see lines **-*).

L54: It may help the reader to understand the concept of mating success better if you actually give an example of how it is calculated

Author response : We have added such example in the new version of the Introduction when we define mating success (lines **-*).

L57: No idea what this limiting case is about

Author response : We have deleted the reference to this extreme scenario (see our answer below).

L57-59: This feels like a straw man as it is almost impossible for all individuals in a population to receive pollen from all donors (unless the population is very small and discrete).

Author response : We agree with AE that this extreme scenario is very unlikely to occur unless there is almost not spatial structure in pollen dispersal, which we know exists. In the previous version, we aimed at describing a theoretical case where the measure of Bateman gradients would be uninformative. In the new version of the Introduction, we have reformulated our explanations without referring to this extreme hypothetical scenario (see also our response to reviewer 1).

L72: given that mating success should be counts of mating events, it does not really make sense that you could calculate it using molecular tools. This creates confusion about what you actually mean by mating success. You need to make sure that this issue is addressed much earlier in the manuscript (I see that it is addressed later). I also think that you cannot call this mating success. It is a surrogate for mating success. I suggest that throughout the ms, you call it what it is and motivate that it is in fact a reasonable surrogate for mating success.

Author response : We completely agree with AE that the way mating success is typically calculated only capture parts of the mating events that actually occurred. Our study shares this shortcoming with many other studies estimating Bateman gradients and we aimed at being critical of this aspect of the Bateman approach. In the former version of the manuscript, we chose to refer to our genetically-based

counts of mating partners as « mating success » to be consistent with the terminology of the Bateman literature. However, we also agree that it would clarify things to adopt a terminology for these mating partners counts closer to what they are. We therefore designate in this new version of the manuscript to mating success as « genetically-based mating success proxy», a terminology defined at the very beginning of the new Introduction section. We also addressed this issue of counting mating events by referring to the AE's recently published new methodology to mark and track pollen grains (Minnaar & Anderson, 2019). This new methodology could be key in identifying whether the classical « genetically-based mating success proxies» are good surrogates for mating success in future studies to come.

Minnaar, C., Anderson, B. (2019) Using quantum dots as pollen labels to track the fates of individual pollen grains. *Methods in Ecology and Evolution*. Doi:10.1111/2041-210X.13155.

L93-95: This was the purpose of our recent review in AoB 2019, Minnaar and Anderson, Plant-pollinator interactions along the pathway to paternity. Despite describing the male fitness pathway through the lens of animal pollinated plants, it has many intersections with your own paper in discussions of sexual selection, bateman's predictions, analogies to broadcast spawning, and identifying the problems associated with using genetic tools for calculating mating success.

Author response : We thank AE for pointing out this very relevant review. We now refer to this review regarding plant sexual selection (lines **-*).

L95: Never good to lead with such a perspective as it forces the reader to go back and find the perspective you are referring to. Just state the perspective

Author response : Corrected, we now provide the perspective in that sentence (line **).

L101: Not sure what you mean by “dam.“ The word is used at least twice in this para

Author response : With the terminology “sires“ and “dams“, we were referring to how are described the male and female parents in the quantitative genetic literature. We agree that it might be confusing to readers and we replaced by “males“ and “females“ throughout the text.

L128, 129: these terms are not properly explained here and because the reader does not know exactly what you mean, it makes the rest hard to follow. I saw that they were explained (in terms of calculations) in the methods. Perhaps it should be done here.

Author response : We have now clarified this point by providing a brief description of these three Bateman metrics at the very beginning of the Introduction so that the reader better understands the concepts the Introduction is centered on (see lines **-*).

L134: I was very surprised to see any hypotheses here. This work is completely hypothesis-driven and I would like to see explicit hypotheses developed here.

Author response : We agree that a clear statement of the hypotheses was lacking in the previous version of the manuscript. It is now added on lines **-*.

L152: It was never made very clear why this experiment is done at 2 different densities. This needs to be clear and it should be worked into the hypotheses what you expect to find by this and why.

Author response : This is a crucial aspect in the paper. We agree with AE that we could expect differences in the strength of sexual selection in plants with variation of plant density. It was actually the aim of our computer simulation to investigate the effect of plant density on sexual selection metrics and to test the hypothesis that the effect of plant density should depend on the scale of pollen dispersal. Therefore, our computer simulations predict that depending on the abilities of males to disperse their pollen, the intensity of sexual selection increases either with an increase or a decrease of density. We agree with AE on the need to clearly state our hypotheses and we have now added

some information about what we expect regarding the effect of plant density on sexual selection in our computer simulations (see lines **-**).

However, we are reluctant to add an hypothesis test comparing our two-common gardens at high- and low-density. Our plant density are not replicated in our experimental design and therefore we cannot rule out that the differences observed between common gardens are not the result of unmeasured differences occurring between them. This is why we did not aim at formally comparing them and rather investigated the effect of plant density using computer simulations. Our efforts to avoid comparing the two gardens was appreciated by reviewer 2 (although he required that we eliminate a paragraph in the Result section, see below). Yet, we believe that including the data we collected from the two common gardens is useful because their plant density cover the range of densities found in natural populations of *Mercurialis annua*. Therefore the fact that differences in Bateman gradients are found in both gardens suggest that our findings regarding sex-roles and sexual selection may be rather general across *Mercurialis* geographical range. We are now clear on the fact that our study does not aim at comparing plant densities experimentally but that the two gardens and their contrasted plant densities provide Bateman gradient at two densities representative of extreme densities commonly found in this species (see lines **-**).

Finally, the two-densities arise because these two common gardens are a subset of a larger experiment aimed at (1) comparing male and female plastic responses to plant density (Tonnabel *et al.* 2017) and (2) using experimental evolution to test whether male morphology evolve with more intense male-male competition triggered by increased plant density (experiment on going). We extracted only one population at low- and high-density because we were limited by the number of genotyping that we were able to perform.

Tonnabel, J., David, P., Pannell, J.R. (2017) Sex-specific strategies of resource allocation to competition for light in a dioecious plant. *Oecologia*, **185**:675-686.

L153: It is also not very clear at this stage why the paired design was important. I think it needs to be motivated more clearly.

Author response : We have added a sentence justifying the importance of the paired design in the last paragraph of the introduction (see line **-**). We also now

L317: what is meant by pollen distance?

Author response : Corrected to « pollen dispersal distance ».

L334, 335: What is meant by non-trivial? It feels meaningless and non-quantitative

Author response : Agreed, we changed this sentence to better summarize the simulation results : « Our simulations revealed opposite effects of plant density on Bateman metrics depending on the pollen distance dispersal ability ».

L451: spatial conformation - Do you mean density?

Author response : Yes, we changed to plant density to clarify the sentence (see line **).

Referee: 1

This manuscript reports on a study of sexual selection in plants using a dioecious, wind-pollinated herb. It shows that, like in animals, that male fitness is more variable than female fitness, with male fitness increasing with increasing access to mates, whereas this is not the case for female fitness. As such, it confirms earlier studies in animals and some studies in plants. Where it differs from previous plant studies is that it takes a Bateman-gradient approach. That said, earlier work on the topic needs to be acknowledged and discussed. The authors are building on a foundation and by not citing this foundational work they are preventing naïve readers from knowledge of a richness of studies approaching the question of what limits male vs. female reproductive success. I am thinking of

studies such as Queller 1983 Nature, Meagher 1986 Am Nat, Schoen 1986 Evolution, Meagher 1991 Am Nat, to name a few. The work presented in the manuscript is well executed, important, fairly well explained, and makes good use of the study system.

Author response : We thank reviewer 1 for this summary of our study and for pointing out these relevant literature. Following advice of reviewer 1 and as described in length in our general response above, we have enriched both the Introduction and Discussion sections with additional references (see the list of added references at the end of our general response above). The changes that we implemented in this new version of the manuscript aimed at better putting our work using the Bateman gradient methodology in perspective with what was done previously on plant sexual selection.

Abstract

Line 9 - the authors use three related phrases: reproductive success, mating success, and mating effort. I don't understand why the last of these is used, and feel that the authors need to define what they mean by reproductive success vs. mating success early on – perhaps in line 20.

Author response : We thank reviewer 1 for pointing out that the manuscript lacked a clear definition of mating and reproductive success. Thanks to reviewer 1's comments, we realized that the manuscript contained inadequate interchange between « mating success » and « mating effort » which led to confusion in the manuscript. As stated above in our general response and in the detailed comments to the AE, we have now defined both «reproductive and mating success » at the very beginning of the Introduction section and we have deleted all references to mating effort as we were indeed referring to number of mates and not to allocation to traits involved in mate acquisition. In the few places where we still discuss the evolution of resource allocation strategies to traits allowing to reach mating partners, we did not use the terminology « mating effort » to avoid introducing a confusion.

Line 10 – should read ‘associated with males’ abilities...’

Author response : Corrected.

Line 12-13 – this is an odd way to end the Abstract. Is this really what the authors want the readers to take away? If so, I do not see that they have sufficiently explained this in the ms.

Author response : We agree with reviewer 1 that the last sentence of the Abstract was perhaps too distant from the actual study and we replaced it by the following statement « Our results suggest that sexual selection acts in a sexually asymmetric manner in a plant species and should shape plant traits allowing pollen to travel further and reach more mates. », (see lines **-**).

Introduction

Line 30 – get rid of mating effort, which is not explained.

Author response : As described above, we have replaced all references to « mating effort » to what we actually meant « mating success ».

Line 42-43 – this attempt to parse things in a way that allows the authors to say they are the first comes across as unnecessary.

Author response : We have rephrased this sentence and more generally reviewed the Introduction section to better highlight how our study articulates in the plant sexual selection literature and what aspects of the study are novel without stating that we are the first to estimate Bateman gradients in flowering plants. We agree that this point is not central in the manuscript. The changes are displayed on lines **-**.

Lines 49-68 – I don't understand what the authors are trying to get at here. What do they mean by ‘limiting case’ and why do they think that anyone would assume equal mating success? This dichotomy seems like a straw man that doesn't help the reader. ‘Notorious’ (defn: famous or well

known, typically for some bad quality or deed) does not seem like a word that should be applied to plants.

Author response : We initially thought that it was useful to describe an extreme unrealistic scenario to illustrate the problems arising from estimating mate numbers thanks to genetically-based paternity analysis performed on a limited number of seeds. Since both reviewer 1 and AE found this reference nonhelpful, we did delete the reference to equal mating. We have now hopefully clarified our rationale about the possibility that we missed mating partners (see lines **-**). This point will also be clarified by the fact that we did change the term designating mating success to *genetically-based mating success proxy* throughout the manuscript.

Lines 89-98 – this paragraph seems misplaced. It does not ‘introduce’ something about the study in a useful way to the reader.

Author response : We agree with reviewer 1 that this paragraph was not very helpful. In the new structure of the Introduction and Discussion sections, we have deleted this paragraph and added a reference to variance partitioning only when we introduce our study in the last paragraph of the Introduction (see lines **-**).

Lines 112-117 – I advise the authors to lose the terms ‘budget’ and ‘direct.’ They simply add an unnecessary layer to the framework. I am not alone in thinking that fecundity selection and sexual selection are not the same thing and should be differentiated from each other (Arnold, Shine, Losos, Blanckenhorn). This paragraph muddles the two.

Author response : We completely agree that it is helpful to decompose selection into components acting on gamete production and on access to mates (i.e. fecundity and sexual selection). In the plant literature, selection acting on plant size through fecundity selection is classically referred to as « budget » effects of plant size and selection acting on plant size through sexual selection as « direct » effect of plant size. It was indeed our aim, as reviewer 1 suggests, to introduce the idea in this paragraph that both types of selection are acting. However, it can happen that both types of selection (i.e. fecundity and sexual) may be muddled together if a trait favors both gamete production and access to mates. That is actually what is suggested by the results of selection gradient estimations in a paper that we recently got accepted in *Evolution* (i.e. some vegetative morphologies that are selected participate both to pollen production and dispersal). Because this paper (Tonnabel *et al.* 2019) is clearly framed within the terminology of « budget » and « direct » effects, we believe it is useful to keep this terminology here too. However, the comment of the reviewer made us realize that our message was not clear and we changed the paragraph to clarify that we were indeed distinguishing selection acting through gamete production or through mate acquisition. In particular, we are now referring to fecundity selection and sexual selection (see lines **-**).

Tonnabel, J., David, P., Klein, E.K., Pannell, J.R. (2019) Sex-specific selection on plant architecture through 'budget' and 'direct' effects in experimental populations of a wind-pollinated herb. *Evolution*, doi.org/10.1111/evo.13714.

Discussion

The Discussion starts well, but again, acknowledging previous work by other authors belongs here.

Author response : As explained in our general response above, we have considerably enriched our references to previous studies on plant sexual selection (see the added reference list at the bottom of this letter). In particular, at the beginning of the Discussion section, we now review studies that have investigated how plant traits may be selected through sexual selection and we discuss previous findings related to sex-differences in reproductive success (see lines **-**).

Line 418 – consider citing De Cauwer *et al.* 2010 *Mole Ecol.*

Author response : This reference is now cited on line **.

Lines 422-432 – unless male plants can judge density (how would that happen? I can see females doing via pollen deposition), this paragraph does not make sense to me.

Author response : In the case of our simulation work, we were not considering that males were able to perceive density necessarily but rather that plant density would select for the male strategies allowing to gain more mates and competitively excluding other males (i.e. either dispersing pollen locally or further away). We are now clearer on the fact that no perception of density mechanism is required for selection of male vegetative architecture influencing their pollen dispersal kernel, and therefore, their access to mating partners (see lines **-**). However, it is well known that overall density is perceived by plants (through modification of light composition after crossing vegetation) and that plants commonly plastically change their resource allocation in response to density. One could imagine that such perception of global plant density (in a situation where sex-ratios are even) could have selected for plastic responses in males to adjust their resource allocation to traits influencing their pollen dispersal (thus optimizing their chances of mate acquisition depending on plant density). We are now clearer that the two possibilities are conceivable (lines **-**).

Lines 443-445 – I appreciate the analogy to broadcast spawners.

The finding that pollen dispersal, but not their measure of the number of pollen grains (pollen weight), affects mating success says to me that this really is sexual selection at work and not merely fecundity selection. The authors might consider adding something to this effect.

Author response : We share reviewer 1's opinion that the independence of pollen production on mating success compared to the clear association of pollen dispersal distance with mating success points to a case of sexual selection. We have now added a reference to that interpretation at the beginning of the Introduction, lines **-**.

Referee: 2

I have reviewed the manuscript entitled “Do metrics of sexual selection conform to Bateman's principles in a wind-pollinated plant?” for publication in Proc B. This MS reports a study in which 100 male plants and 100 female plants were planted in gardens of either low (n=1) and high density (n=1). After a four weeks of mating, all adults and seeds (about 600+ per garden) were genotyped to infer paternity. Authors used these data to estimate metrics of sexual selection, which are widely used in animal biology but have apparently not been applied to plant systems so far. Interestingly, the data suggest that plants show similar patterns of sexual selection than what is generally observed in animals. The authors further decompose the variance observed in male reproductive success along multiple fitness components, contrasting male fitness acquired through a close female plant (planted in the same pot) and the one gained through all the other females in the population. Most of the variance in fitness seems to arise from the mating success gained with the other females in the population. Finally, the authors ran computer simulations to explore the role of plant density on the operation of sexual selection, revealing that whether sexual selection is stronger at low or high density ultimately depends on males' ability to disperse.

In general, I enjoyed reading the MS. I am not an expert in plant biology and, given how widespread Bateman metrics are in animal biology, it strikes me why such a study had not been done before. The MS reads very well, the results are interesting, and the MS leads to interesting follow-up questions about the relationship between sexual selection and plant density. I have a few comments that are mainly minor and meant to be useful.

Author response : We thank reviewer 2 for this summary of our findings. It is indeed interesting that no study to date have applied the batemanian approach to flowering plants. We hope that our study will pave the way for future use of this methodology in plant clades and that the approach will be useful to understand sexual selection operating in plants.

General comments

(1) I appreciate that authors clearly say that the experimental design was not done to compare density because low and high density were not replicated (line 164-166). One cannot compare two treatments that have not been replicated. The authors have successfully refrained themselves from comparing these two treatments, with the exception of the paragraph line 327-332, which should be deleted (along with any other potential comparison I may have missed elsewhere).

Author response : We fully agree with reviewer 2 that we cannot compare the results from the two common gardens as the two densities were not replicated. In the previous version, we clearly stated that our aim was not to compare these two gardens but yet, we were ambiguously doing so in a few sentences. The paragraph pointed by reviewer 2 was indeed not adequate and we deleted it. We have also deleted other parts in the manuscript that were comparing the two common gardens (in the Results and Discussion sections) as well as the corresponding parts in the statistic description for consistency.

(2) The variance decomposition approach is well-suited for such an experimental design with a paired female and extra-paired females. I however feel that more details are required. First, authors say that all covariances were estimated (line 238), but the results show only 5 covariances (Fig. 2). Please clarify why e.g. cov MS-PS were not explored in the intra-pair fitness route? Is it that some cannot be estimated, or are not biologically interesting? Is it that covariances can be estimated only within each route (intra vs inter)? I feel that more details should be given here, and please use consistent abbreviation, e.g. does Pat share = Pat = share? Second, my understanding of variance decomposition approach is that confidence intervals can be estimated through bootstrapping, which would be handy to formally estimate how good each variance value is known and to compare variance components to one another (e.g., Fig 2 in Janicke et al, Am Nat, 2015). I believe this is a necessary addition, as it would allow authors to tell if variances are statistically different from one another which, strictly speaking, cannot be done in the current version of the MS.

Author response : We have added explanations on the calculation of co-variances (lines **-** and in the Supplementary Methods S1). As reviewer 2 pointed out, we do not explore the covariance between mating success and either paternity share or fecundity of mates at the intra-pair scale. This is simply because paternity shares and fecundity can be computed only on those males that have a non-null intrapair mating success : among them, intrapair mating success is a constant (=one, the only female in the pair) so it cannot covary with anything. Similarly, the males that did not mate with any extra-pair female were excluded from the calculations of the co-variances at the extra-pair scale, but this time the number of extrapair female mates can be 1, 2, 3... and can covary with the other parameters. We have reviewed these explanation in supplementary methods S1 and added a sentence in the main text to explain why certain covariances cannot be calculated (see lines **-*). We have also checked for consistency of abbreviations throughout the manuscript. As stated in our general response above, we have now computed 10,000 bootstraps on the calculation of all variance terms. This allows to compare the variance terms by comparing their 95 % CIs following methods described in Janicke *et al.* (2015). This is now explained in the Material and Methods (lines **-*), the results are shown in Figure 2 and explained on lines **-*). We have not compared all variance terms with each other because this was not the aim of our study but for the extra-pair mating success which was the main driver of variance in male reproductive success, we did provide the percentage of bootstrap replicates in which extra-pair mating success yielded the highest variance value (see lines **-*).

Specific comments

156: What is the rationale for raising all plants at low density? Is that possible that the density of young plants will influence adult sexual traits, such as e.g. pollen dispersal traits?

Author response : We have added a rationale for keeping all plants at an initial low density (lines **-*). As reviewer 2 points out, plants are known to be able to adjust their resource allocation strategies to architectural traits (e.g. plant size, branch length). In particular, plants typically increase their growth when confronted to high plant density in a « shade avoidance response ». Even if our primary

aim was not here to compare results from the two common gardens, we wished to avoid introducing a source of variance in plant vegetative traits between common gardens that would simply result from competition for light. Indeed, vegetative traits adjusted in the plastic « shade avoidance » response are also involved in the pollen dispersal (as shown in Tonnabel *et al.* 2019). Therefore, we thought that it was simpler to have an experimental design where our two gardens would solely differ in terms of the density applied when matings were assessed.

Additionally, the two common gardens are a subset of a larger experimental evolution protocol (still ongoing) in which we wished to avoid that plastic response to density would obscure the relationship between a plant's genotype and its morphological traits. Indeed such plastic response to density may have weakened our ability to select on traits involved pollen dispersal (i.e. if most males end up being tall as a result of light competition it could mask males that were tall initially).

179-180: Given that relatively few seeds have been genotyped per female (about 6.5 per female plant), it is likely that sires have been missed in the paternity analyses. How is it supposed to affect the results?

Author response : This is indeed a critic that can be raised to most study on Bateman gradients and that we wished to point out in the manuscript, as our study shares that critic with many others. We have revised the manuscript to make that important point clearer and to review studies that have investigated the potential consequences of missing mates in genetic assays (see lines ****_****, ****_****, ****_**** in the Introduction, Material and Methods and Discussion sections). Importantly, this point is now introduced very early in the introduction (lines ****_****), please see our general answer regarding that point.

334-344: the simulation is a nice part of the MS and, in my opinion, it would be nice to have a figure showing these results.

Author response : We agree with reviewer 2 that the simulation results are an important part of the manuscript. We initially chose to keep these results in Appendices because of space constraint. Following reviewer 2's suggestion, we have now added a small figure (see Fig. 4) showing a simulation corresponding to a random distribution of plants and to variable abilities to disperse pollen as it is the scenario closest to the situation commonly found in natural populations. The new figure 4 displays results for Bateman gradients only as results are anyway consistent with other Bateman metrics. We however decided to keep Fig. S2 as it is as it is useful to see all the results together to understand the main message of the simulations.

Table 1: Does the correction for Binomial sampling error not change the results, or is it a typo?

Author response : This was not a typo (although the rounding of I at high density had not been done correctly), at both gardens, opportunity for selection was equal with or without binomial correction.

Table 2: Are all betas standardised? Some are very large and non-significant (e.g., extra-pair partner RS), while others are tiny and significant (extra-pair PS).

Author response : Only pollen dispersal distance and pollen weight are standardized therefore the betas are expressed in the unity of the response variable (i.e. the fitness component). It is therefore not abnormal that values of betas are on different scales regardless of their significance. We have carefully checked all the statistics and everything was correct. In the case of the extra-pair PS (paternity share) component that reviewer 2 was referring to, it actually reflects an increase of PS of +4% through the range of pollen dispersal distance represented in the data, which is not nothing (given that open average PS is around 16%). We however agree that an important information was lacking in the previous version of the manuscript to properly interpret these betas : we have added in the new version of Table 2 a line showing the mean value in each fitness components.

List of added references :

TO DO